# Dissociation of salts in water under pressure

Cunzhi Zhang [1], Federico Giberti [2], Emre Sevgen[2], Juan J. de Pablo[2,3], Francois Gygi [4] & Giulia Galli [2,3]✉

The investigation of salts in water at extreme conditions is crucial to understanding the properties of aqueous fluids in the Earth. We report first principles (FP) and classical molecular dynamics simulations of NaCl in the dilute limit, at temperatures and pressures relevant to the Earth's upper mantle. Similar to ambient conditions, we observe two meta-stable states of the salt: the contact (CIP) and the solvent-shared ion-pair (SIP), which are entropically and enthalpically favored, respectively. We find that the free energy barrier between the CIP and SIP minima increases at extreme conditions, and that the stability of the CIP is enhanced in FP simulations, consistent with the decrease of the dielectric constant of water. The minimum free energy path between the CIP and SIP becomes smoother at high pressure, and the relative stability of the two configurations is affected by water self-dissociation, which can only be described properly by FP simulations.

[1] Department of Materials Science and Engineering, COE, Peking University, 100871 Beijing, China. [2] University of Chicago, 5640 S. Ellis Ave., Chicago, IL 60637, USA. [3] Materials Science Division, Argonne National Laboratory, Argonne, IL 60439, USA. [4] University of California Davis, Davis, CA 95616, USA. ✉email: gagalli@uchicago.edu

In the Earth's mantle water-rich fluids play a key role in many processes, including metasomatism[1,2], magma production in subduction zones[3] and the Earth's carbon cycle[4,5]. These fluids are saline, with properties dependent on the ionic concentration[6–8]. For example, at high pressure ($P \sim 1$ GPa) and temperature ($T \sim 1000$ K) the presence of $Na^+$ and $Cl^-$ dissolved in water is known to greatly increase the electrical conductivity of aqueous fluids[9–11] while substantially decreasing the activity of water[12,13] and increasing the solubility of several crustal rocks[6,12]. Despite the key role of salts in water in determining the properties of the Earth's mantle constituents, very little is known about the relation between the macroscopic physical properties of saline solutions at high $P - T$ and their molecular structure. In particular, the association of oppositely charged ions (ion-pairing) deserves special attention since it is the first step in the nucleation process of salts, and ultimately affects the conductivity of saline solutions[14–16].

At ambient conditions, ion pairing in water has been investigated by several experimental techniques, including X-ray scattering and X-ray absorption[17], neutron diffraction isotope substitution[18], and dielectric relaxation spectroscopy[19,20]. At high temperature and pressure, where water becomes highly corrosive, only few probes are available. Usually salt solutions are studied by Raman spectroscopy[21,22] and data have so far been reported up to ~700 K and ~1.5 GPa, showing that, similar to ambient conditions, there are two stable ion-pair configurations present in simple salt solutions, such as NaCl: the contact ion-pair (CIP) and the solvent-shared ion-pair (SIP), where the ions are separated by at least one water solvation shell. However experiments are not available at higher pressure.

A number of simulations have been performed at ambient conditions to study the kinetics and thermodynamics of the CIP/SIP conversion[23–28], with focus on NaCl. No such study of ion pairing has yet been reported at high $P$ and $T$, due to the many requirements in simulating solutions at extreme conditions, including a proper description of water dissociation and of the substantial changes in hydrogen bonding relative to ambient conditions[29–31].

Here, we report a computational study of ion pairing in NaCl dissolved in water at conditions relevant to the Earth's mantle, in the dilute limit, corresponding to a concentration similar to that of the average ocean salinity. We carry out free energy calculations using first principles molecular dynamics (FPMD) simulations with the PBE exchange-correlation functional and empirical molecular dynamics (MD) simulations, coupled with enhanced sampling methods. We consider $P = 11$ GPa and $T = 1000$ K, i.e. conditions similar to those at the bottom of the Earth's upper mantle[31,32]. We find key changes of ion-water and ion-ion interactions occurring at extreme conditions, which affect the free energy surface (FES) of the salt in water, in particular the relative stability of contact and separated ion pair configurations, the barrier between them and the minimum free energy path (MFEP) connecting the two metastable states. We also find quantitative differences between results obtained with first principles simulations and empirical force fields, e.g. related to the relative stability of the two metastable states, with first principles calculations allowing us to examine the effect of water self-dissociation on ion-water and ion-ion interactions.

The rest of the paper is organized as follows: we first describe the solvation structure of $Na^+$ and $Cl^-$. We then present our results on free energy calculations for the salt at ambient and extreme conditions, followed by a discussion on entropic and enthalpic contributions to the free energy and on the influence of water dissociation. We close the paper with a discussion section.

## Results

**Solvation shell structure of NaCl in water.** We begin our discussion by describing the structural properties of the solutions, and in particular the solvation shell structure of $Na^+$ and $Cl^-$, a basic property used to understand association and dissociation mechanisms at the microscopic level and hence to interpret the free energy surfaces described below. We discuss in detail our first principles results, obtained with the PBE exchange-correlation functional, pointing out differences with empirical potentials when notable. We note that the PBE functional adopted here has been shown to accurately describe the equation of state[32] and vibrational spectra[31] of water under pressure, as well as Raman cross sections[33], at several high $P - T$ conditions. Importantly the PBE functional yielded predictions of the dielectric constant of the liquid consistent with previous measurements[32].

Our PBE results are shown in Supplementary Table 1 and Fig. 1 for a NaCl solution at ambient and high $P - T$ conditions. The concentration of the two solutions is the same, 0.68 M, and corresponds to a dilute limit (for comparison, the average salinity of the oceans on Earth correspond to a salt concentration of 0.6 M). The first noticeable difference between ambient and extreme conditions is the larger number of water molecules in the solvation shells of $Na^{+}$[34,35] and $Cl^-$ ions at high pressure: the average coordination number for $Na^+$ increases from 4.0 to 6.4 for the CIP state and from 5.2 to 7.6 for the SIP state. For $Cl^-$, the average coordination number increase is more pronounced, from 6.4 to 12.6 for the CIP state and from 6.9 to 12.9 for the SIP state. The trends observed at the PBE level are similar to those found using empirical potentials, however force fields yield a slightly larger average coordination number for both ions (see Supplementary Table 1).

To identify the orientation of water molecules we used the angle between Na–O and Cl–O bonds and the water dipole moment, denoted as $\theta$(Na–O) and $\theta$(Cl–O). Given the small separation of $Na^+$ and $Cl^-$ in some of the configurations, ion solvation shells may be overlapping. Hence, we carried out separate calculations for water molecules belonging to $Na^+$, $Cl^-$ and shared $Na^+$-$Cl^-$ solvation shells. At ambient conditions, the water molecules that are not shared exhibit only one orientation around the ions, corresponding to an angle $\theta$(Na–O) of $\simeq 45°$ for $Na^+$ and $\theta$(Cl–O) of $\simeq 130$ for $Cl^-$, in all three states: CIP, SIP, and TIS (transition state between CIP and SIP). However, shared water molecules behave differently: in the SIP state their orientation closely resembles that of the molecules that are not shared, while in the CIP and TIS states the distribution of angles $\theta$(Na–O) ($\theta$(Cl–O)) is peaked at relatively higher (lower) values.

We found that at high $P - T$, all angular distributions are broadened and become less structured, especially that of the Cl-dipole orientation. We also found that the angle between the Na–O (Cl–O) direction and the water dipole increases (decreases) at extreme conditions, as shown in Supplementary Fig. 6. The changes observed under pressure are more noticeable for shared molecules. The broad distribution found at high $P - T$ is indicative of a higher mobility of water between solvation shells and shared regions of the liquid, which may facilitate dissociation and re-association processes and hence influence free energy surfaces, as discussed below. The trends observed here with FPMD are again similar to those found with most empirical potentials, with some minor quantitative differences. For example the average $\theta$(Na–O) obtained with force fields at high $P - T$ is ~56°, while that for PBE functional is ~61° (see Supplementary Figs. 7 and 8). However as we report in the next section, FPMD and empirical force fields predict different free energy differences between SIP and CIP under pressure, in spite of similarities in solvation shell structure and ion coordination numbers found within the two descriptions of interatomic forces.

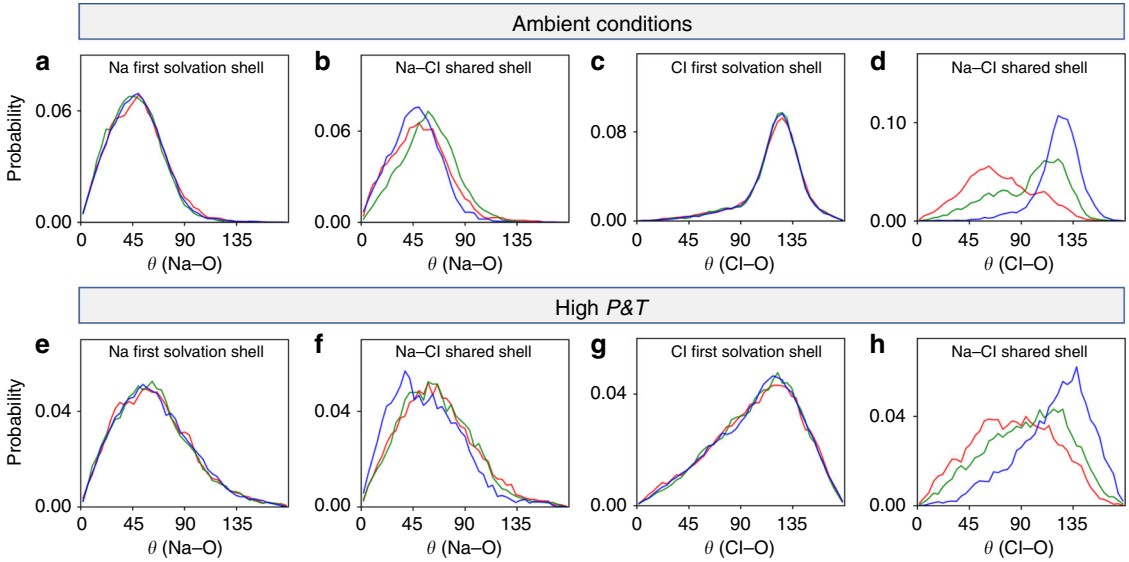

**Fig. 1 Angular distribution functions at ambient and extreme conditions.** Probability distributions of angles ($\theta$) between Na- and Cl-O bonds and the dipole of water molecules belonging to the ion first solvation shell (**a**, **c**, **e**, **g**), and the dipole of water molecules belonging to the shared Na–Cl shell (**b**, **d**, **f**, **h**), obtained with first principles molecular dynamics simulations. **a–d** and **e–h** show the results for CIP (red line), TIS (transition state between CIP and SIP; green line) and SIP (blue line) at 400 K and 1 atm and 1000 K and 11 GPa, respectively. The cutoff value used to define the first solvation shell is 3.2 (3.9) Å for Na$^+$ (Cl$^-$) ion, chosen as the first minima of the ion-O radial distribution function. Since we performed biased simulations (with an external force acting on the ions), we used a weighting scheme to recover unbiased statistics (see Supplementary Note 4).

**Free energy surface of NaCl in water at ambient conditions**. We now turn to the discussion of free energy surfaces, which were computed as a function of ion separation ($r$), as well as both ion separation and cation coordination number ($n$) using the adaptive biasing force (ABF) sampling method[36]. We report in the Supplementary Figs. 1 and 2, a comparison of results obtained with empirical potentials carried out for different concentrations varying from 0.1 to 0.85 M at ambient conditions and from 0.17 to 0.7 M at high pressure. Within these ranges we did not observe any noticeable difference between the computed one- (1D) and two- (2D) dimensional free energy surfaces. Our first principles results described below are obtained for a concentration of 0.68 M and are considered as representative of the entire regime (0.17–0.7 M).

In order to understand the association and dissociation mechanism of Na$^+$ and Cl$^-$ ions, we first examined the 1D FES. At ambient conditions, the free energy surfaces $F(r)$ as a function of the distance between the ions present two minima, CIP ($r \approx 2.8$ Å) and SIP ($4 < r < 5.5$ Å), with a well defined barrier between them, both in first principles and classical MD simulations[23,24] (see Fig. 2a, b). To facilitate the comparison between the different potentials, we report in Table 1 the Na–Cl separation, and the free energy barrier and free energy difference between CIP to SIP. We note that the Na–Cl separation distance in the SIP state is shorter in FPMD than in classical simulations, possibly due to the presence of flexible water molecules which are allowed to dissociate. However both first principles and classical calculations predict that SIP is slightly more stable than CIP, with the exception of the $\omega$B97X functional. Note that at ambient conditions our PBE simulations were conducted at 400 K (instead than 300 K as in reference[23]), so as to obtain a description of structural and diffusion properties in better agreement with experiments, as indicated by many previous studies[37–39]. From Fig. 2a, b and Table 1 we can identify two classes of empirical potentials: the SPCE_JC family that favors the SIP stability over CIP more markedly than first principles PBE results; and the TIP3P_JC, RPOL_SD and SPCE_SD force fields, which yield instead results closer to the PBE ones. The comparison between

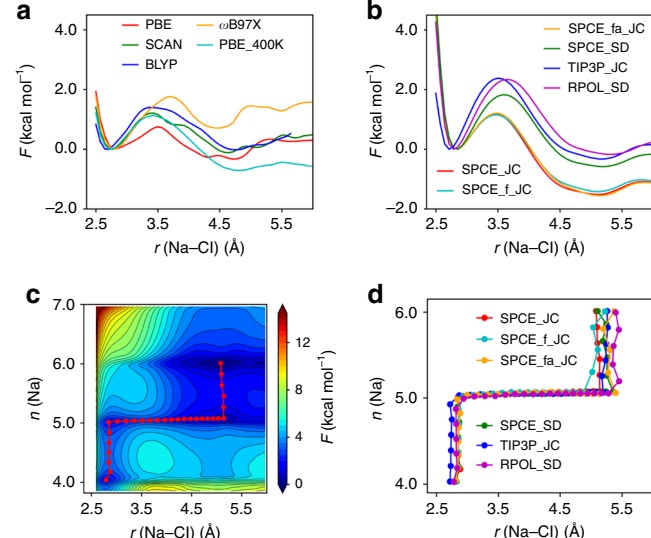

**Fig. 2 Free energy surfaces (FESs) at ambient conditions.** **a** 1D FESs as a function of Na-Cl distance $r$ obtained from first principles, using four different exchange-correlation energy functionals: PBE/SCAN/$\omega$B97X[23], BLYP[24] at $T = 300$ K and the PBE functional at 400 K. **b** 1D FESs obtained with different empirical potentials. The first principles results at 300 K are from refs. [23,24] all other results are from this work. **c** 2D FES as a function of $r$ and $n$ (Na-water coordination number) obtained with the SPCE_JC potential. The minimum free energy path (MFEP) is indicated by red dots. **d** MFEPs obtained with different empirical potentials.

SPCE_SD (non-polarizable) and RPOL_SD (polarizable) seems to indicate that the inclusion of polarization tilts the balance in favor or the CIP minimum, although we note that polarization is not the only difference between the two empirical potentials.

Next, to understand in detail the effect of the solvent[40,41] we computed the 2D FESs $F(r, n)$ by considering the Na–Cl distance and as a second collective variable (CV) the Na–water

**Table 1 Free energy properties.**

| Model | M moles l$^{-1}$ | CIP Å | TIS Å | SIP Å | $F_{TIS} - F_{CIP}$ kcal mol$^{-1}$ | $F_{SIP} - F_{CIP}$ kcal mol$^{-1}$ |
|---|---|---|---|---|---|---|
| PBE[a] | 0.68 | 2.73 | 3.42 | 4.75 | 1.15 ± 0.32 | −0.70 ± 0.49 |
| PBE[23] | 0.85 | 2.82 | 3.52 | 4.76 | 0.75 ± 0.05 | −0.26 ± 0.07 |
| BLYP[24] | 0.85 | 2.68 | 3.38 | 4.74 | 1.4 ± 0.28 | −0.03 ± 0.07 |
| SCAN[23] | 0.85 | 2.74 | 3.44 | 4.60 | 1.22 ± 0.05 | −0.11 ± 0.06 |
| $\omega$B97X[23] | 0.85 | 2.82 | 3.68 | 4.44 | 1.77 ± 0.04 | 0.69 ± 0.06 |
| revPBE-D3[42] | 0.85 | 2.98 | 3.75 | 4.68 | 0.75 ± 0.08 | −0.41 ± 0.08 |
| SPCE_JC[a] | 0.43 | 2.83 | 3.45 | 5.11 | 1.20 | −1.50 |
| SPCE_f_JC[a] | 0.43 | 2.83 | 3.45 | 5.10 | 1.18 | −1.40 |
| SPCE_SD[a] | 0.43 | 2.83 | 3.59 | 5.18 | 1.86 | −0.55 |
| TIP3P_JC[a] | 0.43 | 2.70 | 3.52 | 5.17 | 2.40 | −0.32 |
| RPOL_SD[a] | 0.43 | 2.77 | 3.65 | 5.31 | 2.35 | −0.16 |

Positions of free energy minima (contact ions pair—CIP, transition state—TIS, separate ion pair—SIP) are given in the third to fifth column and free energy differences in the last two columns. The model is specified in the first column, in terms of either the functional used in density functional theory calculations, or the empirical potential. The concentration of the solution (M) is shown in the second column. All data are at 300 K and 1 atm.
[a]This work. PBE results are obtained at 400 K and 1 atm. The results are from 1D FESs with the free energies of CIP, TIS, and SIP denoted as $F_{CIP}$, $F_{TIS}$ and $F_{SIP}$, respectively.

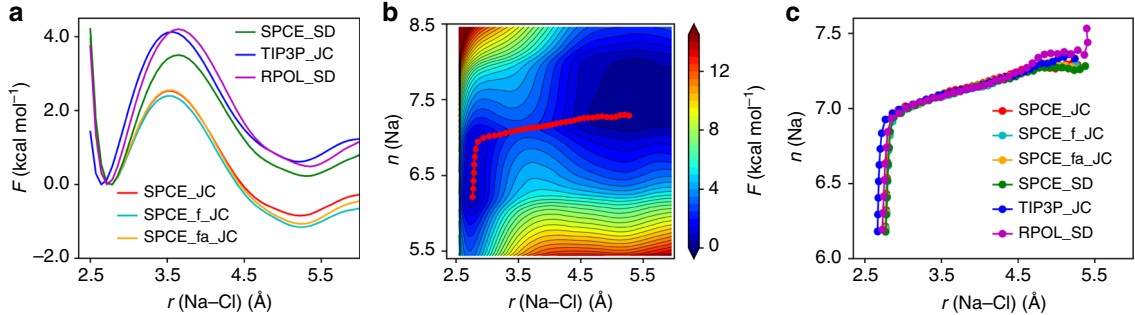

**Fig. 3 Free energy surfaces (FESs) at high pressure and temperature.** Results obtained with empirical potentials for 0.68 M solutions. Calculated 1D FESs (**a**), 2D FES (**b**, obtained from SPCE_JC with minimum free energy path (MFEP) shown by the red dots), and MFEPs (**c**) obtained with different force fields.

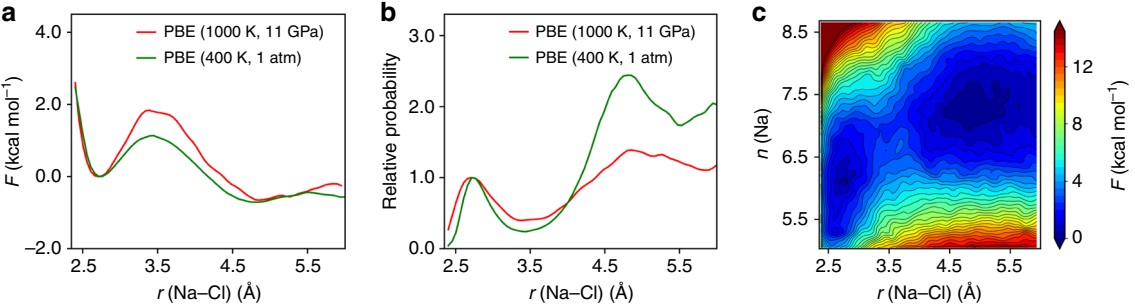

**Fig. 4 Free energy surfaces (FESs) at high pressure and temperature.** Results obtained with first principles simulations. Calculated 1D FESs (**a**), probability distribution of ion-pair (**b**), and 2D FES (**c**). The probability distributions are aligned at the position of the contact ion pair (CIP): $\exp(-(F(r) - F_{CIP})/k_B T)$. $F$ is the free energy, $T$ the temperature, $r$ the Na–Cl distance and $k_B$ the Boltzmann constant.

coordination number ($n$, Eq. (1) in the "Methods"), and the corresponding minimum free energy pathway with the Nudged Elastic Band method. Our results are reported in Fig. 2c, d for the SPCE_JC potential (the results of other force fields are given in Supplementary Figure 9 and they are qualitatively similar). Interestingly, we found that the 2D FES is more corrugated than the 1D FES, with several metastable states involved in the dissociation process. The MFEPs in Fig. 2d illustrate that the dissociation of Na$^+$-Cl$^-$ from CIP to SIP occurs by first increasing the water–Na$^+$ coordination, and then separating the ions, which eventually leads to a further increase of the ionic

coordination. We will see below that such corrugation disappears almost entirely at extreme conditions, leading to a much different dissociation path.

**Free energy surface of NaCl in water at extreme conditions.** Increasing $T$ and $P$ has a substantial effect on the salt solvation properties. The results for free energy surfaces obtained using force-fields and FPMD with the PBE functional are reported in Fig. 3 and Fig. 4, respectively. The relative stability and the height of the barrier are reported in Table 2. We found that the barrier

**Table 2 Free energy properties.**

| Model | M moles l$^{-1}$ | CIP Å | TIS Å | SIP Å | $F_{TIS} - F_{CIP}$ kcal mol$^{-1}$ | $F_{SIP} - F_{CIP}$ kcal mol$^{-1}$ |
|---|---|---|---|---|---|---|
| SPCE_JC | 0.68 | 2.77 | 3.54 | 5.23 | 2.55 | −0.84 |
| SPCE_f_JC | 0.68 | 2.78 | 3.54 | 5.23 | 2.41 | −1.15 |
| SPCE_SD | 0.68 | 2.74 | 3.65 | 5.30 | 3.56 | 0.28 |
| TIP3P_JC | 0.68 | 2.65 | 3.55 | 5.23 | 4.14 | 0.61 |
| RPOL_SD | 0.68 | 2.72 | 3.63 | 5.37 | 4.21 | 0.49 |
| PBE | 0.68 | 2.71 | 3.37 | 4.84 | 1.84 ± 0.29 | −0.65 ± 0.45 |
| PBE_r | 0.68 | 2.70 | 3.37 | 5.21 | 1.46 ± 0.29 | −1.26 ± 0.45 |

Positions of free energy minima (contact ions pair—CIP, transition state—TIS, separate ion pair—SIP) are given in the third to fifth column and free energy differences in the last two columns. The model is specified in the first column, in terms of either the functional used in density functional theory calculations, or the empirical potential. The concentration of the solution (M) is given in the second column. All data are at 1000 K and 11 GPa.
The results are from 1D FESs with free energies of CIP, TIS, and SIP denoted as $F_{CIP}$, $F_{TIS}$, and $F_{SIP}$, respectively.
PBE_r denotes simulations where the O–H bonds of the water molecules were kept fixed to 1.0 Å.

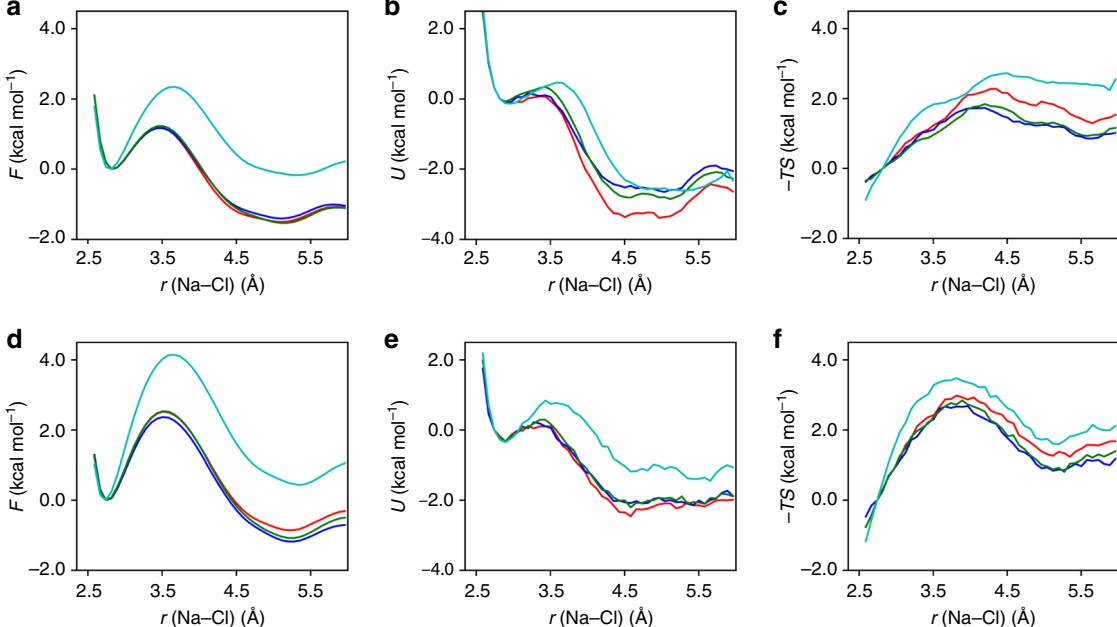

**Fig. 5 Enthalpic (_U_) and entropic (_−TS_) contributions to free energy surfaces (_F_). a–c** and **d–f** show results for NaCl solutions at concentration 0.43 M at ambient conditions and 0.68 M at 1000 K and 11 GPa, respectively. We report results obtained with four different empirical potentials : SPCE_JC (red line), SPCE_f_JC (blue line), SPCE_fa_JC (green line) and RPOL_SD (cyan line).

required for the salt dissociation is substantially higher than at ambient conditions. In addition, the several metastable states between CIP and SIP present at ambient conditions, are absent at high $T$ and $P$ (Figs. 3b, c and 4c). The observed broad path between the two states indicates that dissociation may occur with Na$^+$ in different coordination configurations, while at ambient conditions the dissociation path is confined to a narrow valley, corresponding to specific configurations and coordination numbers. This finding hints at the possibility of different solvation properties of solid surfaces in contact with saline solutions at the conditions of the Earth's mantle, relative to ambient conditions.

There are important, quantitative differences between the results of empirical and FPMD simulations. At (1000 K, 11 GPa), the PBE results in Fig. 4a show a larger free energy barrier for NaCl ion-pair dissociation from CIP to SIP, of ∼1.84 kcal mol$^{-1}$, than at ambient conditions, similar to the predictions of empirical potential simulations, as illustrated in Supplementary Fig. 11 and Fig. 3a, which can again be categorized into two families, similar

to ambient conditions. However the free energy barrier is larger with empirical potentials, as shown in Tables 1 and 2. Importantly, the results of some empirical potentials (SPCE_SD, TIP3P_JC and RPOL_SD) suggest that the CIP minimum becomes more stable at extreme conditions (Supplementary Fig. 12 and Fig. 3a), while FPMD predicts that the SIP remains the most stable minimum, although slightly less stable than at ambient conditions. This decreased stability of the SIP at extreme conditions is clearly reflected in Fig. 4b, where we report the ion-pair distribution probability, which shows an increased stability of the CIP. Given the ability of the functional used here (PBE) to describe the equation of state, dielectric constant[32] and vibrational spectra[31] of water under pressure, we consider the results obtained from first principles as the most accurate ones, and they may be used to validate empirical potentials for future studies.

Results obtained using force fields (Supplementary Fig. 10 and Fig. 3b) and PBE (Fig. 4c) for 2D free energy surfaces are found to display similar features, with only two metastable states (CIP/SIP)

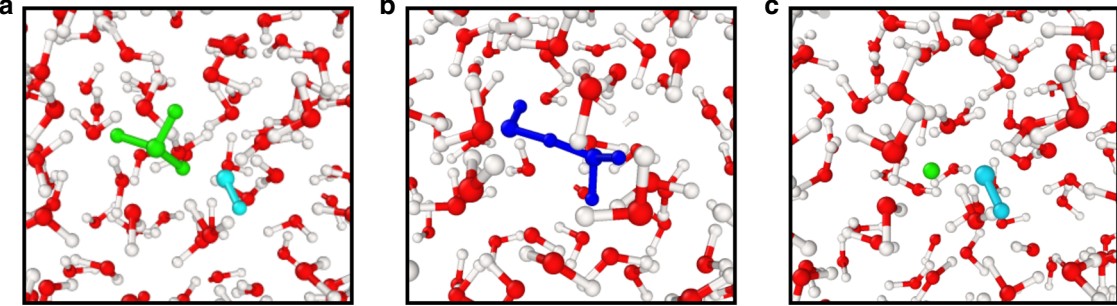

**Fig. 6 Snapshots of high pressure simulations.** Snapshots extracted from first principles simulation of a 0.68 M NaCl solution at 1000 K and 11 GPa, showing short-lived $OH_3^+$ and $OH^-$ (**a**), $H_4O_2$ (**b**) and $OH^-$ and $H^+$ (**c**), shown with cyan, green and blue spheres, respectively.

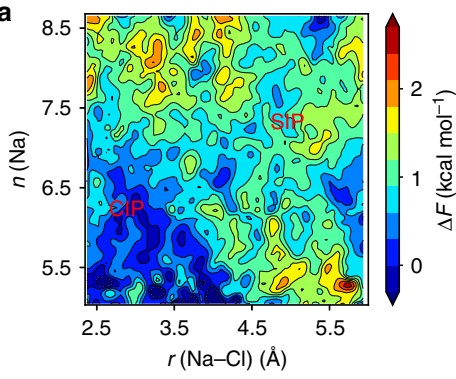

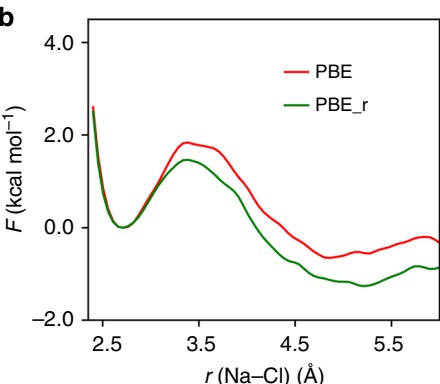

**Fig. 7 Impacts of water dissociation on free energy surfaces. a** 2D free energy difference, $F_{PBE} - F_{PBE\_r}$, obtained using ab initio simulations of a 0.68 M NaCl solution at 1000 K and 11 GPa with the PBE functional. PBE_r denotes FPMD the results with O–H bond fixed to 1.0 Å and the positions of CIP and SIP minimum are marked in red. **b** 1D FESs obtained with (green line) and without (red line) rigid molecules aligned by CIP state.

and a broader dissociation path than at ambient conditions, likely due to the change in the solvation shells discussed earlier.

The increased stability of the CIP at extreme conditions is consistent with the decrease of the water dielectric constant found in our previous study[32]. Given that at the conditions considered here, the water dielectric constant is less than one half of that at ambient conditions, the moderate decrease in stability of the SIP is somehow surprising and the change in dielectric properties seems to affect more the dissociation path between CIP and SIP than their relative stability. Although further studies as a function of $P$ and $T$ are necessary to draw definitive conclusions, a decrease of the dielectric constant may be expected to lead to a decrease of water activity[43] in deep fluids, relative to a solution at the same concentration at ambient conditions.

**Enthalpic and entropic contributions**. To obtain a deeper understanding of NaCl dissociation free energy profiles, we decomposed the free energy ($F$) into entropic ($-TS$) and enthalpic ($H = U + PV$) components as presented in Fig. 5. The enthalpic term $H$ is calculated as an average potential energy, since the system volume $V$ is constant in our MD simulations, while the entropic term ($-TS$) is obtained as $-TS = F - H$. The convergence of classical potential energy surfaces with empirical potentials required ~400 ns of simulation time. Unfortunately, such decomposition is not feasible at present using FPMD and we rely here on the analysis of classical trajectories only.

Under both ambient and high $P - T$ conditions, the SIP is found to be enthalpically favored (Fig. 5b, e) when using both an empirical polarizable potential and force fields of the SPC_JC family, due to stronger ion-water interactions, while the CIP is entropically favored (Fig. 5c, f), as some tightly solvated water molecules are released when $Na^+$ and $Cl^-$ approach each other, leading to an entropy gain[27,28,44,45]. To understand the effect of the water model, we compared simulations with non-flexible (SPCE_JC), flexible (SPCE_f_JC and SPCE_fa_JC) and polariable (RPOL_SD) force-fields. The FESs of the flexible and non-flexible models are very similar as well as their enthalpic and entropic contributions; the FESs obtained with the polarizable force field are quantitatively different, however they yield the same results in term of enthalpic and entropic contributions, indicating that the identification of these contributions is a robust result.

Relative to ambient conditions, at high $T$ and $P$ the CIP/SIP enthalpy difference decreases, reducing the free energy difference between the two states, while a larger entropic contribution is responsible for an increase of the dissociation/association barrier between the CIP/SIP state.

**Influence of water dissociation**. The importance of water dissociation can only be assessed by using our first principles results. Using PBE trajectories we characterized the speciation of water in the saline solution and investigated its possible effects on NaCl dissociation.

In our FPMD simulations, water dissociation events can be directly observed, with the creation of short-lived charged species such as hydronium and hydroxide. Some examples of the configurations found in our simulations are shown in Fig. 6. To identify these transient chemical species, we adopted a geometrical criterion where the cutoff distance for the O–H bond is chosen to be 1.25 Å, corresponding approximately to the first minimum of the OH radial distribution function[31]. For each species, we characterized its probability distribution and lifetime. We observed the formation of $OH^-$, $OH_3^+$ and $H_4O_2$ species[29–31]. We found that the results obtained for pure and saline water exhibit the same distribution probability. All these species are found to have a relatively short lifetime (<10 fs) indicating that

dissociation and re-association of water molecules occur very rapidly at high pressure and temperature[31].

In order to evaluate whether water dissociation has a substantial impact on the salt dissociation process, we performed an additional biased FPMD simulation of 300 ps in which the O–H bonds of each water molecule were constrained to be 1.0 Å, hence preventing the formation of any hydronium and hydroxide or any other ions (The H–O–H angle of water molecules was not constrained). The results indicate that the effects of water dissociation on angular distribution for solvated water molecules are minor (see Supplementary Fig. 5). In Fig. 7a, we present the difference between the FESs obtained with rigid and flexible water simulations. Fig. 7b shows that constraining the O–H bond length increases the stability of SIP over that of the CIP (by ~0.6 kcal mol$^{-1}$). This result is consistent with our findings of entropic contributions being responsible for the CIP stabilization: water dissociation is expected to increase entropic contributions and hence to stabilize CIP against SIP. The entropic contribution coming from dissociation is clearly absent in all empirical potentials, which nevertheless show an excessive stabilization of CIP relative to SIP, relative to first principles MD, most likely due to parameters of the force field being fitted at conditions too far from those simulated here.

## Discussion

We reported a study of NaCl in water under pressure, at high temperature, at conditions relevant to the Earth's upper mantle (1000 K, 11 GPa). We carried out molecular dynamics simulations coupled with enhanced sampling techniques, using both first principles calculations based on density functional theory (DFT) and empirical force fields. Similar to ambient conditions we found the presence of two metastable states in the solution at high $P$ and $T$: a contact ion pair (CIP) which we found to be entropically favoured, and a separated ion pair (SIP), which is instead enthalpically favoured. Both at ambient and extreme conditions the SIP is more stable than the CIP. However we observed a slight decrease in the stability of the SIP relative to ambient conditions when using first principles simulations, consistent with the decrease of the water dielectric constant under pressure, found in our previous study[32]. With both force-fields (polarizable and non-polarizable) and simulations based on DFT we found a higher barrier separating CIP and SIP. We note however that although all simulations predict an enhanced stability of CIP relative to SIP under pressure, such enhancement is rather moderate in our first principles MD simulations, while it is much more substantial with most force fields. These differences between first principles and empirical simulations are not really surprising as none of the existing force fields was fitted to data obtained under extreme conditions. Given the ability of the density functional used here (PBE) to describe the equation of state[32] and vibrational spectra[31] of water under pressure, we consider the results obtained from first principles as accurate. In addition, we found that at extreme conditions the dissociation path of the salt differs from that observed at ambient conditions, in particular no intermediate states corresponding to different solvation structures are observed. Finally, our results showed that the Na$^+$–Cl$^-$ association is influenced by rapid water self-dissociation events, which lower the free energy difference between SIP and CIP, with the former minimum remaining however the most stable. Our simulations represent the first study of the free energies of salt in water under pressure and lay the foundation to understand the properties and influence of salts dissolved in water at pressure and temperature relevant to the earth mantle, where salt-rich fluids are believed to control the chemistry of fluid-rock interaction. In particular, the notable decrease of the dielectric constant will influence the interaction of saline solutions with solid surfaces. Such interaction will also be different than at ambient conditions due to the greatly modied dissociation path found at extreme conditions.

## Methods

**First Principles molecular dynamics simulations.** First principles molecular dynamics simulations (FPMD) were carried out using the Qbox code[46] coupled with the SSAGES suite of codes in client-server mode[47,48]. We used the PBE functional[49], plane wave basis sets and optimized norm-conserving Vanderbilt (ONCV) pseudopotentials[50,51], with an energy cutoff of 60 Ry to expand the electronic wave functions[52]. We modeled a NaCl solution of concentration ~0.68 M using a periodic cubic cell of volume (13.46 Å)$^3$ containing 126 (80) water molecules and 1 NaCl formula unit at 1000 K and 11 GPa (400 K and 1 atm). The elevated temperature of 400 K at low pressure was adopted to improve accuracy in the description of the water structure at ambient conditions[37–39] Deuterium was used instead of hydrogen to allow for a larger MD timestep of 0.24 fs, and 0.68 fs for flexible and rigid water simulations, respectively.

**Classical MD simulations.** Classical MD simulations were performed using the LAMMPS package[53] coupled with the SSAGES suite of codes, and several force fields. We used the SPCE[54], SPCE_f/SPCE_fa[55] (SPCE_f: flexible water model; SPCE_fa: O–H bond constrained water model), TIP3P[56] and RPOL[57,58] (polarizable). To represent the Na$^+$ and Cl$^-$ ions we compared the results obtained with two different force fields, obtained with the Joung and Cheatham[59] (JC) and Smith and Dang[58] (SD) parameters for the Lennard–Jones potential, respectively. At (1000 K, 11 GPa), the simulation box size was determined so as to obtain a water density of 1.57 g cm$^{-3}$, i.e. equal to the density in our FPMD simulations. At (300 K, 1 atm), the size of the simulation cell was determined using NPT simulations. The time-step adopted in all classical simulations is 0.25 fs. For consistency with our FPMD simulations, deuterium, instead of hydrogen, was used also in simulations with force-fields.

**Enhanced sampling.** Two collective variables (CVs) were used in our calculations, the Na–Cl distance, $r$ (Na–Cl), and the coordination number of Na$^+$ (number of water molecules in its first solvation shell), $n$ (Na), which is defined using the continuous and differentiable function:

$$n = \sum_i \frac{1 - \left(\frac{r_i - d_0}{r_0}\right)^a}{1 - \left(\frac{r_i - d_0}{r_0}\right)^b} \qquad (1)$$

where $r_0 = 1.0$ Å, $d_0 = 2.2$ Å, $a = 4$, $b = 12$, and $r_i$ is the distance between Na$^+$ and oxygen of the $i$th water molecule. We chose these two CVs as they have been previously identified to be crucial in the ion pairing process in water[40,41]. The adaptive biasing force[36] (ABF) method was chosen to drive the dissociation and association of the ions. In ABF, a grid is defined in the CVs space and the derivative of the FES is estimated at each time step, so as to counter-balance the forces experienced by the atoms during the simulations. A discrete grid of 40 × 40 was used for the $r$ and $n$ variables to store the local estimates of the mean derivative of FES.

To speed up FPMD simulations we took advantage of the multiple walkers capability implemented in SSAGES and used 15 walkers, for a total FPMD simulation time of ~300 ps (450 ps) for the NaCl solution at high (ambient) $P - T$ conditions. At high $P - T$ we also repeated a 300 ps simulations for water with rigid O–H bonds. In classical simulations we employed a single walker for a total of 50 ns simulation time. The convergence of FES with respect to system size and simulation time was tested with classical MD simulations (see Supplementary Figs. 1–3). At high pressure, in FPMD simulations the 1D potential of mean force was obtained from the 2D FES by integrating the coordination degree of freedom and properly normalizing it (see Supplementary Note 4). The error bar of our FPMD results in Tables 1 and 2 are estimated based on simulations of equivalent simulation time using SPCE_JC potential (Supplementary Note 3).

## Data availability

Data are available at: https://paperstack.uchicago.edu/qrespexplorer and https://paperstack.uchicago.edu/paperdetails/5eb387ba1c8f784c6043d6b3?servers=https%3A%2F%2Fpaperstack.uchicago.edu.

## Code availability

Qbox, SSAGES and LAMMPS are free and open-source codes available at http://qboxcode.org, https://ssagesproject.github.io and https://lammps.sandia.gov, respectively. Data processing scripts are available upon request from the authors.

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

## Acknowledgements

This work was supported by the Midwest Integrated Center for Computational Materials (MICCoM), as part of the Computational Materials Sciences Program funded by the U.S. Department of Energy, Office of Science, Basic Energy Sciences, Materials Sciences and Engineering. This work was completed using resources provided by the University of Chicago's Research Computing Center. C.Z. acknowledges the China Scholarship Council (CSC) for sponsoring his visit to the University of Chicago. We thank Marivi

Fernandez-Serra, Viktor Rozsa, Hythem Sidky, Dimitri Sverjenski and Jonathan Whitmer for many useful discussions.

## Author contributions

F.Giberti, F.Gygi, J.J.d.P. and G.G. designed the research. C.Z. performed the calculations. E.S. contributed to the method for free energy calculation. All authors contributed to the analysis and discussion of the data and the writing of the paper.

## Competing interests

The authors declare no competing interests.
