## [Peer Review File · Nature Communications]

Reviewers' comments:

Reviewer #1 (Remarks to the Author):

In this work Galli and co-workers perform a series of ab initio molecular dynamics simulations of NaCl under pressure comparing the results of classical potentials to those from first principles. They find the existence of contact minima and solvent separated minima in the potentials of mean force and that this behavior is present in both ambient pressure and high pressure simulations. There are apparently some discrepancies between the stability of the contact vs solvent separated minima in the force fields compared to ab initio.

Overall, this is an interesting contribution. However, I am not so convinced that there is any big surprise in these set of calculations that would warrant publication in Nature Communications. The part that I think is most interesting is the role of water dissociation on the $\text{Na}^+ \cdots \text{Cl}^-$ dissociation. This is however buried right at the end of the paper and would require significant work to bring that part of the paper to the forefront. In particular:

i) Figure 7 left panel differs significantly from Figure 2 bottom left. The question is why? It seems like water dissociation is adding barriers and creating minima in all the regions of free energy space that weren't there before.

ii) Related to the water dissociation and its role, it's interesting to speculate a bit about the role of pH on NaCl nucleation and dissociation. Is this known/understood? Any experiments on this front?

iii) Again in reference to the role of water dissociation, an examination of how the enthalpic/entropic contributions come from the dissociation and how they tune the total free energetics would be extremely interesting and would take this paper from being a more specialised one to something with broader impact.

Reviewer #2 (Remarks to the Author):

The first principles MD (FPMD) study of salts under high P and T has been expertly performed and a welcome addition to the literature. However, the accompanying classical MD studies attempt to serve multiple purposes, raise more questions than providing answers, and overall distract from the main message given by the FPMD results. The first issue with the classical force fields is that they are optimized at ambient conditions and should not be used at high P and T. There are several studies showing their failure away from ambient conditions. The main reason for this failure is the lack of polarization. Curiously, polarization was completely ignored in the force fields used in this study. Instead the effect of flexibility of water molecules was considered, which is rather negligible compared to polarization. Polarization has been incorporated in some force fields. So if the authors are keen to compare the FPMD results with classical ones, they better use such force fields. I also note that such comparisons carried out at different temperatures (i.e. 300 vs 400 K) have no clout, as the results differ substantially between the two temperatures. If there is a problem with the accuracy of water molecules at 300 K (that is the excuse for using 400 K instead of 300 K in the FPMD simulations), it could also happen at 1000 K and 11 GPa.

I recommend using the classical MD results only to address sampling issues, separating free energy contributions, etc, but not for comparison with the FPMD results. To give an example, the difference between the results using the SPCE and TIP3P water models is even larger than those affected by increases in T and P, which would make anyone very suspicious of the relevance of such diverse results from the force fields in the context of ab initio simulations.

Reviewer #3 (Remarks to the Author):

The authors present classical and first-principles (ab initio, with PBE functional) molecular dynamics simulations of NaCl-H₂O solutions at extreme conditions with the objective of understanding the nature of how Na and Cl ion associate. To this end, they map-out the free energy surfaces for NaCl ion association and the relative energies of the contact ion pair and the solvent-separated ion pair.

To understand the solvation of Na and Cl, they present angular distributions of the solvent molecules about the Na and Cl (Figure 1). Perhaps I'm missing the point, but I don't understand what the angular (vs. the radial) distribution is supposed to tell us. More fundamentally, I don't understand why they do not present Na-Cl radial distribution functions from unconstrained simulations using both classical and ab initio methods to illustrate the presence of the two complexes and whether the classical potentials agree with the first-principles results. This seems like an obvious omission. Classical (SPCE/SD) unconstrained simulations (e.g., Sherman and Collings, *Geochemical Transactions*, 3, 102-107, 2002) show that the solvent-separated NaCl ion pair disappears anyway at high temperature (325 °C), as would be expected. I would have thought that perhaps the effect of pressure might enhance the formation of SSIP even at high T, but apparently not.

Although the presentation of the paper could be improved, I cannot find any technical issues of concern. The main problem is simply that the objectives of the paper and the result of the study are not significant enough to warrant publication in *Nature*. It is not clear why we should care about the relative free energies of solvent-separated ion pairs vs contact ion pairs; we already knew such ion pairs exist. The study did not yield results that were especially unexpected and the implications of the results to our understanding of the upper mantle is unclear. As it is, therefore, I think this paper is of too-limited interest. Perhaps the authors could (given the free energy of the two types of complexes) calculate the resulting electrical conductivity and effective ionic strength of NaCl-H₂O solutions under mantle conditions. That would at least give the results a bit more impact and put the results in a geochemical/geophysical context. Still, I think even that would not really warrant publication in *Nature*. If the authors actually calculated the activity of water as a function of NaCl concentration, P and T and then was able to argue that a free NaCl-H₂O fluid could (or could not) coexist with mantle minerals at high P,T, they would have a *Nature* paper!

Response to reviewers

We thank the referees for their suggestions and comments. We have accordingly revised our manuscript by taking into account all their concerns. Following is a summary of our revision, as well as a point-by-point reply to all the comments.

Reviewer #1

Overall, this is an interesting contribution.

We thank the reviewer for the positive comment, and we address the reviewer's concerns and suggestions below.

However, I am not so convinced that there is any big surprise in these set of calculations that would warrant publication in Nature Communications. The part that I think is most interesting is the role of water dissociation on the Na^+ - Cl^- dissociation. This is however buried right at the end of the paper and would require significant work to bring that part of the paper to the forefront.

We believe the following results reported in our paper could not be anticipated and they have important implications:

- We found a decreased stability of the SIP relative to the CIP at extreme conditions. While this result is consistent with our previous results showing a decrease of the water dielectric constant at these conditions, it is interesting to see that the change in stability between the two minima is rather moderate, given the change of more than a factor of 2 in the dielectric constant.

We added a discussion on this point:

Page 13: [...] The increased stability of the CIP at extreme conditions is consistent with the decrease of the water dielectric constant found in our previous study³². Given that at the conditions considered here, the water dielectric constant is less than one half of that at ambient conditions, the moderate decrease in stability of the SIP is somehow surprising and the change in dielectric properties seems to affect more the dissociation path between CIP and SIP than their relative stability. Although further studies as a function of P and T are necessary to draw definitive conclusions, a decrease of the dielectric constant may be expected to lead to a decrease of the water activity in deep fluids, relative to a solution at the same concentration at ambient conditions, and in turn to influence the serpentinization of certain rocks as well as the kinetics of chemical reactions. [...]

- We found a notable increase in the barrier between the two minima (CIP and SIP) at extreme conditions: such a barrier is expected to influence the kinetic of chemical reactions at the interface of saline solutions and solid surfaces, e.g. rocks.
- We found a drastic change in the dissociation path relative to ambient conditions, which is again expected to impact the solvation properties of rocks, as well as chemical reactions at interfaces, as it is now noted on page 13 and in the discussion session:

Page 18: [...] the notable decrease of the dielectric constant will influence the interaction of saline solutions with solid surfaces. Such interaction will also be different than at ambient conditions due to the greatly modified dissociation path found at extreme conditions. In addition, the increased barrier between contact and separate ion pairs will influence the kinetic of chemical reactions in the presence of saline solutions at solid surface.

We feel our results on water self-dissociation are better discussed after presenting the free-energy surfaces and entropic and enthalpic contributions. Following the referee's suggestion, we added a discussion to this part:

Page 17: [...] Our results show that constraining the O-H bond length raises the stability of SIP over that of the CIP (by ~ 0.6 Kcal/mol). This result is consistent with our findings of entropic contributions being responsible for the CIP stabilization: water dissociation is expected to increase entropic contributions and hence to stabilize CIP against SIP. The entropic contribution coming from dissociation is clearly absent in all empirical potentials, which nevertheless show an excessive stabilization of CIP relative to SIP, relative to first principles MD, most likely due to parameters of the force fields being fitted at conditions too far from those simulated here. [...]

In particular:

i) Figure 7 left panel differs significantly from Figure 2 bottom left. The question is why? It seems like water dissociation is adding barriers and creating minima in all the regions of free energy space that weren't there before.

We apologize for the confusion regarding Fig.7, caused by an incorrect labeling of the figure which has now been corrected. As specified in the caption, the left-hand side of Fig. 7 is a difference of free energies obtained with dissociative and rigid water molecules. It is now labeled ΔG (and not G, as erroneously done in the original manuscript).

ii) Related to the water dissociation and its role, it's interesting to speculate a bit about the role of pH on NaCl nucleation and dissociation. Is this known/understood? Any experiments on this front?

This is an interesting question that unfortunately we are not yet in a position to answer. Indirectly, our results on the difference in dissociative paths and relative stability of CIP and SIP between dissociative and rigid water simulation point at the important role of pH. However quantifying such an effect is beyond reach of the present days simulation tools (as it would have to be done completely from first principles).

iii) Again in reference to the role of water dissociation, an examination of how the enthalpic/entropic contributions come from the dissociation and how they tune the total free energetics would be extremely interesting and would take this paper from being a more specialised one to something with broader impact.

We have added a discussion on page 17 (reported above) and also a brief sentence to the abstract:

We also found that the relative stability of CIP and SIP is affected by water self-dissociation, which can only be described properly by using a first principles description. In particular the stability of the CIP appears to be enhanced in the presence of dissociation, consistent with the fact that contact ion pairs are entropically favored.

Reviewer #2

The first principles MD (FPMD) study of salts under high P and T has been expertly performed and a welcome addition to the literature.

We thank the reviewer for the positive comments, and we address the reviewer's concerns and suggestions below.

However, the accompanying classical MD studies attempt to serve multiple purposes, raise more questions than providing answers, and overall distract from the main message given by the FPMD results. The first issue with the classical force fields is that they are optimized at ambient conditions and should not be used at high P and T. There are several studies showing their failure away from ambient conditions.

We have rephrased several parts of the paper to clarify that we consider our first principles MD results to be the accurate ones. In particular:

Page 4: [...] These differences between first principles and empirical simulations are not really surprising as none of the existing force fields was fitted to data obtained under extreme conditions. [...]

Page 12: [...] Given the ability of the functional used here (PBE) to describe the equation of state³² dielectric constant and vibrational spectra³¹ of water under pressure, we consider the results obtained from first principles as the most accurate ones, and they may be used to validate empirical potentials for future studies. [...]

The main reason for this failure is the lack of polarization. Curiously, polarization was completely ignored in the force fields used in this study. Instead the effect of flexibility of water molecules was considered, which is rather negligible compared to polarization. Polarization has been incorporated in some force fields. So if the authors are keen to compare the FPMD results with classical ones, they better use such force fields.

We thank the reviewer for the suggestion. We have carried out additional simulations using a polarizable force field and we have added to the revised manuscript the results of simulations carried out both at ambient and extreme conditions (see revised Fig.2,3,5 and revised Table 1 and 2). We comment on the effect of polarization on page 9 of the revised manuscript:

[...] From Fig. 2 and Table 1 we can identify two classes of empirical potentials: the SPCE_JC family that favors the SIP stability over CIP more markedly than first principles PBE results; and the TIP3P_JC, RPOL_SD and SPCE_SD force fields, which yield instead results closer to the PBE ones. The comparison between SPCE_SD (non-polarizable) and RPOL_SD (polarizable) seems to indicate that the inclusion of polarization tilts the balance in favor of the CIP minimum, although we note that polarization is not the only difference between the two empirical potentials. [...]

I also note that such comparisons carried out at different temperatures (i.e. 300 vs 400 K) have no clout, as the results differ substantially between the two temperatures. If there is a problem with the accuracy of water molecules at 300 K (that is the excuse for using 400 K instead of 300 K in the FPMD simulations), it could also happen at 1000 K and 11 GPa.

We thank the reviewer for the remark, which prompted us to include additional explanations to the revised manuscript. Based on the results of several groups in the last 15 years, it is now known that simulations at an elevated temperature, at ambient conditions are necessary to reproduce measured partial distribution functions and diffusion coefficients, when using the PBE functional. Although we recognize this is only a partially satisfactory fix, it is the only one affordable in simulations of the length and complexity of the ones conducted here to obtain free energies. At high P and T the PBE functionals performs instead much better than at ambient conditions, as reported in several of our previous studies of structural, vibrational and dielectric properties of water, and recently also of Raman cross sections (all studies are properly referenced in the text).

In the revised manuscript we write:

Page 3-4: [...] We note that the PBE functional adopted here (see Method section) has been shown to accurately describe the equation of state³² and vibrational spectra³¹ of water under pressure, as well as Raman cross sections, at several high P and T conditions. Importantly the PBE functional yielded predictions of the dielectric constant of the liquid consistent with previous measurements. [...]

Page 12 (also reported above): [...] Given the ability of the functional used here (PBE) to describe the equation of state³² dielectric constant and vibrational spectra³¹ of water under pressure, we consider the results obtained from first principles as the most accurate ones, and they may be used to validate empirical potentials for future studies. [...]

Page 19: [...] The elevated temperature of 400 K at low pressure was adopted to improve accuracy in the description of the water structure at ambient condition, as indicated by many previous studies^{35,48,49}. [...]

I recommend using the classical MD results only to address sampling issues, separating free energy contributions, etc, but not for comparison with the FPMD results. To give an example, the difference between the results using the SPCE and TIP3P water models is even larger than those affected by increases in T and P, which would make anyone very suspicious of the relevance of such diverse results from the force fields in the context of ab initio simulations.

We followed the referee's suggestion and pointed out the differences between different families of empirical potentials (see our response above), emphasizing those closer to our FPMD results and clarifying that the results on sampling obtained with empirical potentials appear to be robust, as the same trends are reproduced by all force fields.

Reviewer #3

To understand the solvation of Na and Cl, they present angular distributions of the solvent molecules about the Na and Cl (Figure 1). Perhaps I'm missing the point, but I don't understand what the angular (vs. the radial) distribution is supposed to tell us.

We have clarified in the text that understanding the structure of solvation shells is important to understand the dissociation path reported later in the paper and this is the reason why they are discussed in detail.

Page 5: [...] We begin our discussion by describing the structural properties of the solutions, and in particular the solvation shell structure of Na⁺ and Cl⁻, a basic property useful to understand association and dissociation mechanisms at the microscopic level and hence to interpret the free energy surfaces described below.

[...]

More fundamentally, I don't understand why they do not present Na-Cl radial distribution functions from unconstrained simulations using both classical and ab initio methods to illustrate the presence of the two complexes and whether the classical potentials agree with the first-principles results. This seems like an obvious omission.

We would like to emphasize that comparisons between first principle and empirical MD results and between different empirical potentials are extensively discussed in our paper, using free energy calculations. Radial distribution functions (RDFs) would give us exactly the same information, including the position of the CIP and SIP minima and their relative stability (there is a direct relation between RDFs and 1D free energy surfaces). One important result of our paper is the calculation of 2D free energy surfaces which tell us, among other information, about the solvation structure changes during the dissociation, which cannot be obtained from RDFs and 1D free energy surfaces.

Classical (SPCE/SD) unconstrained simulations (e.g., Sherman and Collings, Geochemical Transactions, 3, 102-107, 2002) show that the solvent-separated NaCl ion pair disappears anyway at high temperature (325 °C), as would be expected. I would have thought that perhaps the effect of pressure might enhance the formation of SSIP even at high T, but apparently not.

The referee is correct that our results are not obvious at all, and that the combined effect of P and T is complex and gives rise to unexpected results.

Although the presentation of the paper could be improved, I cannot find any technical issues of concern. The main problem is simply that the objectives of the paper and the result of the study are not significant enough to warrant publication in Nature. It is not clear why we should care about the relative free energies of solvent-separated ion pairs vs contact ion pairs; we already knew such ion pairs exist.

It is certainly true that we already knew that such ion pairs exist. However, we did not know their relative stability and the barrier between them at extreme conditions, and how the dissociation path differs at high P and T from that at ambient conditions. As noted above by the reviewer, based on existing results, it was not possible to correctly predict what the combined effect of pressure and temperature would be.

The reasons why we should care is because barriers and dissociation paths will influence chemical reactions at rock surfaces and their kinetics, and we showed here that they are remarkably different than at ambient conditions.

The study did not yield results that were especially unexpected and the implications of the results to our understanding of the upper mantle is unclear.

As reported also in the response to reviewer #1, we believe the following results reported in our paper could not be anticipated and they have important implications:

- We found a decreased stability of the SIP relative to the CIP at extreme conditions. While this result is consistent with our previous results showing a decrease of the water dielectric constant at these conditions, it is interesting to see that the change in stability between the two minima is rather moderate, given the change of more than a factor of 2 in the dielectric constant.

We added a discussion on this point:

Page 13: [...] The increased stability of the CIP at extreme conditions is consistent with the decrease of the water dielectric constant found in our previous study³². Given that at the conditions considered here, the water dielectric constant is less than one half of that at ambient conditions, the moderate decrease in stability of the SIP is somehow surprising and the change in dielectric properties seems to affect more the dissociation path between CIP and SIP than their relative stability. Although further studies as a function of P and T are necessary to draw definitive conclusions, a decrease of the dielectric constant may be expected to lead to a decrease of the water activity in deep fluids, relative to a solution at the

same concentration at ambient conditions, and in turn to influence the serpentinization of certain rocks as well as the kinetics of chemical reactions. [...]

- We found a notable increase in the barrier between the two minima (CIP and SIP) at extreme conditions: such a barrier is expected to influence the kinetic of chemical reactions at the interface of saline solutions and solid surfaces, e.g. rocks.
- We found a drastic change in the dissociation path relative to ambient conditions, which is again expected to impact the solvation properties of rocks, as well as chemical reactions at interfaces, as it is now noted on page 13 and in the discussion session:

Page 18: [...] In particular, the notable decrease of the dielectric constant will influence the interaction of saline solutions with solid surfaces. Such interaction will also be different than at ambient conditions due to the greatly modified dissociation path found at extreme conditions. In addition, the increased barrier between contact and separate ion pairs will influence the kinetic of chemical reactions in the presence of saline solutions at solid surface.

Perhaps the authors could (given the free energy of the two types of complexes) calculate the resulting electrical conductivity and effective ionic strength of NaCl-H₂O solutions under mantle conditions.

We have computed the ionic conductivity of salt solutions under pressure. Our results show a notable increase relative to those of pure water at the same conditions; the conductivity of pure water had been reported in Rozsa et al. PNAS 2018. However, as discussed in the PNAS paper, at present first principles calculations can give us the order of magnitude of ionic conductivities, or in the best-case scenario, conductivities accurate within a factor of 5. Unfortunately, such an accuracy does not allow us to discuss the effect of ion dissociation on the value of the conductivity. Hence, we have chosen not to discuss ionic conductivities in our paper.

That would at least give the results a bit more impact and put the results in a geochemical/geophysical context.

Still, I think even that would not really warrant publication in Nature. If the authors actually calculated the activity of water as a function of NaCl concentration, P and T and then was able to argue that a free NaCl-H₂O fluid could (or could not) coexist with mantle minerals at high P,T, they would have a Nature paper!

The suggestions of the referee are clearly very interesting. However unfortunately the calculations requested are at present unfeasible and we believe that any FPMD practitioner would agree with us on this. In order to answer the referee's questions, we should carry out extensive first principles simulations as a function of concentration and then of the saline solutions at interfaces, amounting to many nanoseconds of simulations for prohibitively large systems (when rocks are included). As we note in the last part of the discussion, ours is a foundational study and unfortunately not all answers are available at present:

Our simulations represent the first study of the free energies of salt in water under pressure and lay the foundation to understand the properties and influence of salts dissolved in water at pressure and temperature relevant to the earth mantle, where salt-rich fluids are believed to control the chemistry of fluid-rock interaction. In particular, the notable decrease of the dielectric constant will influence the interaction of saline solutions with solid surfaces. Such interaction will also be different than at ambient conditions due to the greatly modified dissociation path found at extreme conditions. In addition, the increased barrier between contact and separate ion pairs will influence the kinetic of chemical reactions in the presence of saline solutions at solid surface.

REVIEWERS' COMMENTS:

Reviewer #1 (Remarks to the Author):

I am now happy with all the revisions the Authors have made and recommend for publication.

Reviewer #2 (Remarks to the Author):

The authors addressed all the comments in my report satisfactorily. I recommend acceptance of the manuscript.

Reviewer #3 (Remarks to the Author):

This paper using ab initio MD to explore the nature of NaCl solutions under the conditions of the upper mantle. One of the fundamental questions about such solutions is whether Na and Cl ions have increased association because of the decreasing dielectric constant of water at high P,T. The primary result of this paper is that, at upper mantle PT conditions, the free energy barrier between solvent-separated ion pairs (Na-H₂O-Cl) and contact ion pairs (NaCl) increases. Two of the reviewers suggested that this result is not significant enough to warrant publication in Nature. In response to this point, the authors suggest that the increasing barrier between the two modes of ion association would affect the kinetics of rock-water reactions (e.g., serpentinisation). However, this is not a valid argument: the free barrier between contact vs. solvent-separated ion pairs would not at all define the rate-limiting step in the kinetics of any rock-water reaction. As it is, the increasing free energy barrier between contact and solvent-separated ion pairs does not have any obvious geochemical or geophysical significance.

In summary, I would still conclude that, although the paper is scientifically and technically sound, the results are not of significance to warrant publication in Nature. However, that is an editorial decision and, perhaps the paper must be put into the context of what else has been currently submitted to the journal.